# Cryo-EM structures of Uba7 reveal the molecular basis for ISG15 activation and E1-E2 thioester transfer

Mohammad Afsar[1], GuanQun Liu[2], Lijia Jia[1], Eliza A. Ruben[1], Digant Nayak[1], Zuberwasim Sayyad[2], Priscila dos Santos Bury[1], Kristin E. Cano[1], Anindita Nayak[1], Xiang Ru Zhao[1], Ankita Shukla[1], Patrick Sung[1], Elizabeth V. Wasmuth[1], Michaela U. Gack[2] & Shaun K. Olsen[1] ✉

ISG15 plays a crucial role in the innate immune response and has been well-studied due to its antiviral activity and regulation of signal transduction, apoptosis, and autophagy. ISG15 is a ubiquitin-like protein that is activated by an E1 enzyme (Uba7) and transferred to a cognate E2 enzyme (UBE2L6) to form a UBE2L6-ISG15 intermediate that functions with E3 ligases that catalyze conjugation of ISG15 to target proteins. Despite its biological importance, the molecular basis by which Uba7 catalyzes ISG15 activation and transfer to UBE2L6 is unknown as there is no available structure of Uba7. Here, we present cryo-EM structures of human Uba7 in complex with UBE2L6, ISG15 adenylate, and ISG15 thioester intermediate that are poised for catalysis of Uba7-UBE2L6-ISG15 thioester transfer. Our structures reveal a unique overall architecture of the complex compared to structures from the ubiquitin conjugation pathway, particularly with respect to the location of ISG15 thioester intermediate. Our structures also illuminate the molecular basis for Uba7 activities and for its exquisite specificity for ISG15 and UBE2L6. Altogether, our structural, biochemical, and human cell-based data provide significant insights into the functions of Uba7, UBE2L6, and ISG15 in cells.

Interferon Stimulating Gene 15 (ISG15) was first identified as an interferon (IFN) responsive gene[1]. A large number of studies have focused on delineating the basis of IFN-induced expression of ISG15[2,3], which clearly fulfils an important role in the antiviral response mounted by host cells. Specifically, upon pathogenic challenge, host cells activate various signaling pathways leading to the release of type I IFNs and other pro-inflammatory cytokines[4]. Binding of type I IFNs to their receptor further leads to expression of a number of downstream proteins including ISG15[5]. These IFN-induced proteins play a critical role in host cell defense against viral and other microbial infections. Importantly, a number of studies have been conducted in mouse models where it was observed that *Isg15* KO mice were more

susceptible to viral infections[6]. ISG15 can be conjugated to specific viral proteins, blocking their respective functions. Moreover, host innate immune proteins can undergo ISGylation, prompting their activation. A recent study has shown that ISG15 conjugation is crucial for antiviral response mediated by the viral RNA sensor MDA5, promoting its oligomerization to activate innate immunity against viruses like coronaviruses, flaviviruses, and picornaviruses, while the papain-like protease (PLpro) of SARS-CoV-2 antagonizes ISG15-dependent MDA5 activation through direct de-ISGylation[7]. The de-ISGylating activity of PLpro from SARS-CoV-1 and 2 provides a mechanism to evade the host immune response has been well-established and makes PLpro an attractive drug target[8–12]. Notably, recent studies showed that the

[1]Department of Biochemistry & Structural Biology, University of Texas Health Science Center at San Antonio, San Antonio, TX 78229, USA. [2]Florida Research and Innovation Center, Cleveland Clinic, Port Saint Lucie, FL 34987, USA. ✉e-mail: olsens@uthscsa.edu

balance of conjugated vs. unconjugated intracellular ISG15 determines secretion of extracellular ISG15, which can act 'cytokine-like' and contribute to overzealous inflammation in the host organism[13,14]. Overall, ISG15 plays a significant role in antiviral immunity with several studies highlighting its immunomodulatory function.

ISG15 is a 17.8 kDa protein consisting of two Ubiquitin-like (Ubl) domains[15]. The N-terminal and C-terminal domains of ISG15 share 27 and 37% sequence similarity with Ubiquitin (Ub), respectively[16], and, structurally, both domains are clearly related to Ub. Although, ISG15 was first identified in 1979, its Ubl property was not recognized until 1987[17]. That ISG15 harbors the diglycine motif suggests that its biological role entails its covalent conjugation to lysine residues of target proteins, a process known as ISGylation. Similar to ubiquitination and other ubiquitin-like modifiers, ISGylation requires the activity of a cascade of enzymes comprising E1, E2 and E3[18]. Interestingly, during viral infection, actively translating viral proteins are ISGylated aiding in host response[19–21]. Because they control distinct biological processes, fidelity in the parallel E1/E2/E3 cascades for Ub and the seven related Ub-like proteins in vertebrates including ISG15 is crucial. E1 occupies the apex of the Ub/Ubl conjugation cascade where it performs two main functions: (1) specific activation of Ub/Ubl in a two-step process that results in the formation of a thioester bond between Ub/Ubl and an E1 catalytic cysteine (adenylation and thioester bond formation) and (2) recruitment of cognate E2 enzymes followed by transfer of Ub/Ubl to an E2 catalytic cysteine (thioester transfer). Finally, the Ub/Ubl is covalently attached to specific lysine residues on the substrate via action of the E3 ligase.

While Uba7 has evaded structural characterization, a number of structural studies provide evidence that E1 enzymes comprise multiple functional domains[22–32]. Ub/Ubl is generally recruited via the active and inactive adenylation domains (AAD and IAD, respectively) which are also responsible for Ub/Ubl activation via adenylation. The first catalytic cysteine half domain (FCCH) recognizes Ub/Ubl while the second catalytic cysteine half domain (SCCH), harboring the catalytic cysteine, is responsible for thioester formation with Ub/Ubl (E1-Ub/Ubl). Several studies have shown that E1s undergo large conformational changes that drive Ub/Ubl activation and thioester bond formation[24,25,32,33]. In the case of Uba1, adenylation occurs with the E1 in an "open" conformation while thioester bond formation is accompanied by a major rotation of the SCCH domain to bring the catalytic cysteine into active site (termed the "closed" conformation). Finally, E2 is recognized by the UFD domain followed by passing of Ub from the E1 to E2 catalytic cysteine. While Uba1 (E1) is specific to Ub activation, ISG15 is exclusively activated by Uba7 (E1) and specifically transferred to the E2 enzyme UBE2L6. Although sequence conservation between Uba1 and Uba7 predicts similar domain architecture, in the absence of structural insight, the molecular mechanisms of ISG15 activation, Uba7-UBE2L6-ISG15 thioester transfer, and the rules governing specificity of these interactions remain unknown.

Here, we present cryo-EM structures of human Uba7 in complex with UBE2L6, ISG15 adenylate, and ISG15 thioester intermediate in a conformational state that is poised for catalysis of Uba7-UBE2L6-ISG15 thioester transfer. The structures provide insights into the molecular rules governing specificity of Uba7 for ISG15 and UBE2L6 over Ubls and E2s that function with other canonical E1s. The structures also reveal unique conformations of the ISG15-adenylate NTD and ISG15 thioester intermediate CTD that promote the active conformation of the Uba7-UBE2L6-ISG15(t)/ISG15(a) complex. We present extensive biochemical analysis of our structures which reveal the key determinants of Uba7-catalyzed ISG15 activation and thioester transfer to UBE2L6. Further, we show that these key molecular determinants are crucial for ISG15 function in cells as mutation leads to reductions of global and MDA5-specific ISGylation in human cell-based studies. Altogether, our structural, biochemical, and human cell-based data provide significant insights into the functions of Uba7, UBE2L6, and ISG15 in biology.

## Results

### Trapping doubly-loaded Uba7-UBE2L6-ISG15(t)/ISG15(a) mimetic

To gain insights into the molecular basis by which human Uba7 activates and transfers ISG15 to UBE2L6, we sought to determine the structure of a 'doubly-loaded' Uba7-UBE2L6-ISG15(t)/ISG15(a) mimetic complex. Structural studies of related doubly-loaded E1-E2-Ubl(t)/Ubl(a) complexes for other Ubl systems have historically been hampered by the labile nature of the thioester bond, the low affinity of E1-E2 complexes, and the transience of the intermediate formed during E1-E2 thioester transfer. We recently developed a two-step strategy that overcomes these challenges[27], which we have applied to the ISG15 system here (Fig. 1a). To address the lability of the thioester bond, we utilized a variation of a commonly used approach[34] in which a residue in close proximity to the UBE2L6 catalytic cysteine (Cys86) is mutated to a lysine (L121K) to facilitate ISG15 conjugation to form a stable mimetic of the UBE2L6-ISG15 thioester intermediate, hereafter referred to as UBE2L6-ISG15(t) (Fig. 1a; Supplementary Fig. 1a). To overcome the low affinity of Uba7/UBE2L6 interaction and transience of the intermediate formed during thioester transfer, the catalytic cysteine of the resulting UBE2L6-ISG15(t) thioester mimetic (Cys86) was subsequently crosslinked to the catalytic cysteine of Uba7 (Cys599) (Fig. 1a; Supplementary Fig. 1b). The resulting singly-loaded Uba7-UBE2L6-ISG15(t) represents a mimetic of the Uba7-UBE2L6-ISG15 intermediate during E1-E2 thioester transfer[27,30]. Previous studies in the Ub system have shown that although the single-loaded E1-E2-Ub(t) complex is capable of catalyzing thioester transfer, that double-loaded E1 is significantly more efficient at E2 recruitment and catalysis[35]. To generate a mimetic of double-loaded Uba7-UBE2L6-ISG15(t)/ISG15(a) complex, free ISG15 and ATP/Mg$^{2+}$ were mixed with purified singly-loaded Uba7-UBE2L6-ISG15(t) which allowed binding of ISG15 to the adenylation active site and adenylation of its C-terminal glycine, thus incorporating the ISG15 adenylate (ISG15(a)) into the complex (Fig. 1a; Supplementary Fig. 1c).

### Overall structure of double-loaded Uba7/UBE2L6/ISG15

The double-loaded Uba7-UBE2L6-ISG15(t)/ISG15(a) complex was prepared as described above and imaged by single-particle cryo-electron microscopy (cryo-EM). Two reconstructions of the Uba7-UBE2L6-ISG15(t)/ISG15(a) complex were subsequently isolated, which we refer to as Form 1 and Form 2, with overall nominal resolutions of 3.38 and 3.21 Å, respectively (masked at 0.143 FSC) (Supplementary Fig. 2; Supplementary Table 1). Both reconstructions exhibit the same overall architecture and do not appear to be functionally different, with the main difference between them being the strength of ordering of the SCCH domain, ISG15(a) NTD, and ISG15(t) CTD (Supplementary Fig. 3). Hence, we constructed a composite map and the refined coordinates derived from this map were used for descriptions of the structure below (Fig. 1b and c; Supplementary Fig. 2).

Globally, the Uba7-UBE2L6-ISG15(t)/ISG15(a) complex exhibits a conserved architecture in both forms of the complex with the AAD, IAD, and FCCH of Uba7 engaging ISG15(a) and UBE2L6 sandwiched between the UFD and SCCH of Uba7 (Fig. 1b and c). Cryo-EM density linking Gly157 of the CTD of ISG15(t) to UBE2L6 L121K and linking UBE2L6 Cys86 and Uba7 Cys599 are clearly visible and only slight structural rearrangements would place the relevant atoms in position for catalysis, validating our strategy for trapping the complex (Supplementary Fig. 3a). Molecular recognition of UBE2L6 is combinatorial and involves contacts to the UFD, the SCCH domain, and a tripartite interaction with the crossover loop of Uba7 and ISG15(a) (Fig. 1c cartoon model). A total of ~1400 Å$^2$ of UBE2L6 surface area (from 8400 Å$^2$ total) is buried in the complex. The majority of Uba7 and UBE2L6 are well-ordered (Supplementary Fig. 3b), with local resolutions ranging from 2.8–3.6 Å. By contrast, the SCCH domain is comparatively poorly-ordered with local resolution ranging between 4.0 and 5.8 Å

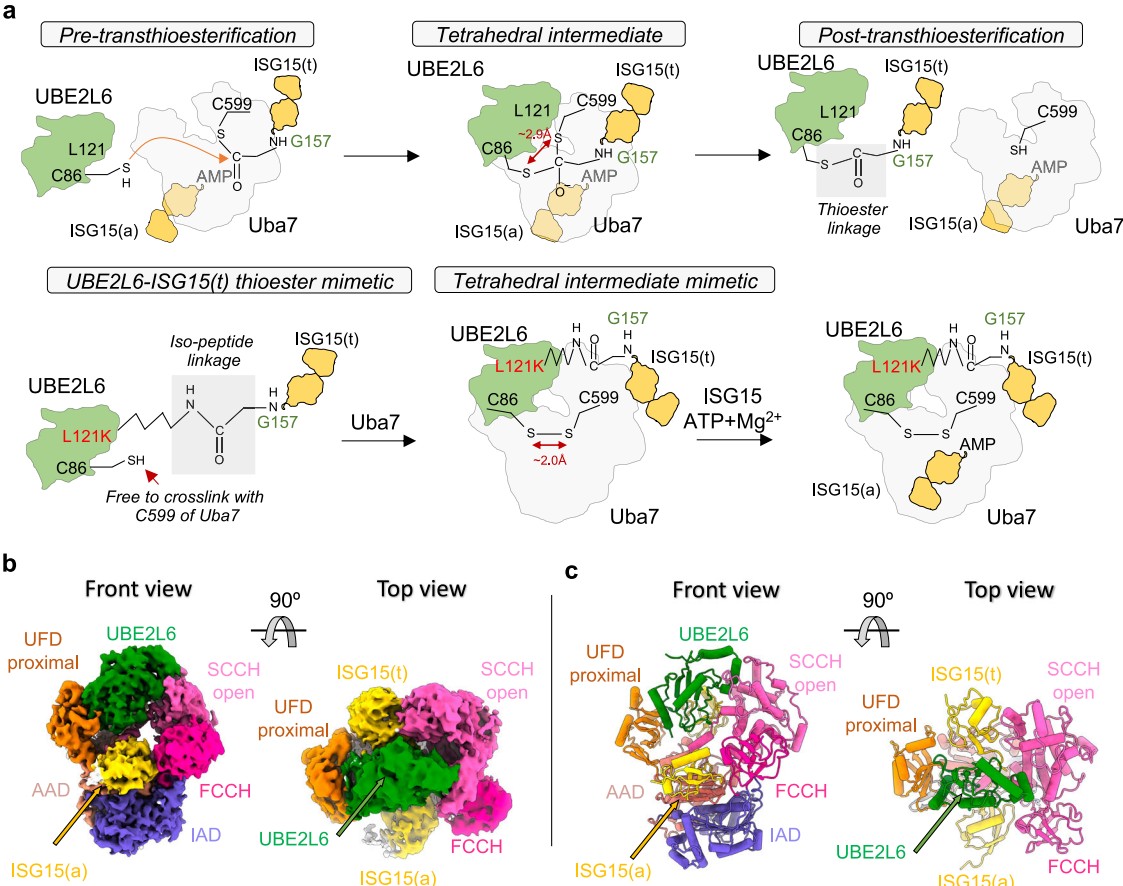

**Fig. 1 | Overall structure of a double-loaded Uba7-UBE2L6-ISG15(t)/ISG15(a) complex. a** Schematic representation of the bona fide Uba7-UBE2L6-ISG15 thioester transfer reaction (*top*). Schematic representation of the chemical crosslinking strategy used to trap a double-loaded Uba7-UBE2L6-ISG15(t)/ISG15(a) complex poised for thioester transfer (*bottom*). **b** Cryo-EM composite map of Uba7-UBE2L6-ISG15(t)/ISG15(a) complex. The IAD and AAD domains of Uba7 are depicted in slate blue and dark salmon, respectively. SCCH domain is displayed in pink, FCCH domain is displayed in deep pink, and UFD domain is displayed in orange. UBE2L6 is depicted in green, while ISG15(a) at adenylation and ISG15(t) at conjugate sites are depicted in gold. **c** Ribbon representation of the Uba7-UBE2L6-ISG15(t)/ISG15(a) complex, presented as in *panel b*.

(Supplementary Fig. 3b). We surmise that the poor ordering of the SCCH domain is likely due to conformational flexibility as discussed further below. Lastly, studies of canonical E1 enzymes for Ub (Uba1 and Uba6)[22,23,31,32], SUMO (SAE)[36,37], and Nedd8 (NAE)[38,39], revealed that the SCCH domain adopts a 'closed' conformation that accompanies thioester bond formation and an 'open' conformation that accompanies adenylation and E1-E2 thioester transfer. These studies also revealed that the UFD domain adopts distal conformations important for E2 recruitment and a proximal conformation that is crucial for E1-E2 thioester transfer by bringing the two active sites into proximity. Consistent with the Uba7-UBE2L6-ISG15(t)/ISG15(a) complex being poised for thioester transfer, the SCCH domain of Uba7 is observed in the open conformation and the UFD adopts the proximal conformation (Fig. 1b and c).

With respect to ISG15(a), the *C*-terminal Ub-like domain (CTD) is well-ordered with a total of ~1500 Å² of ISG15(a) CTD surface area (from ~5200 Å² total) buried in the complex. There is continuous cryo-EM density extending from the C-terminal glycine (Gly157), consistent with the ISG15-adenylate product (Supplementary Fig. 3c). Density corresponding to the pyrophosphate (PPi) leaving group and Mg²⁺ was absent indicating that we have captured the product complex of adenylation, after PPi/Mg²⁺ release. While the local resolution is very modest, the NTD can clearly be seen extending towards the UFD where a crosstalk between the ISG15(a) NTD and the Uba7 UFD appear to take place (Supplementary Fig. 3d). 3D variability analysis of the Uba7-UBE2L6-ISG15(t)/ISG15(a) complex also shows crosstalk between the

UFD and NTD of ISG15(a) as shown in Supplementary Movie 1. Although not included in the final model, the ISG15(a) NTD can be docked into this cryo-EM density and we will discuss the implications of this observation below.

While the density is modest, likely due to conformational dynamism (local resolution of 5–7 Å), the ISG15(t) CTD clearly extends away from the surface of UBE2L6 and is wedged between the UFD and SCCH domain of Uba7 (Fig. 1b, c, top view). Interestingly, while ISG15(a) is located on the 'front' side of the complex in our structure as well as those of other canonical E1/Ubl(a) complexes, ISG15(t) is located on the 'back' side of the complex (Fig. 1b, c top view). Positioning of ISG15(t) on the back side of the complex appears to be a unique feature of the ISG15 system, as Ubl(t) in only other available 'doubly loaded' E1 structures of Uba1 and NAE are located on the front of the complex[27,40]. The sequential events involved in ISG15 adenylation, thiolation, and thioester transfer to UBE2L6 are illustrated in Supplementary Movie 2. The implications and molecular basis for this observation will be discussed in greater detail below. Finally, the NTD of ISG15(t) is not visible in either form of the complex and thus excluded from both models.

## The Uba7 active site
As noted above, the Uba7-UBE2L6-ISG15(t)/ISG15(a) complex harbors the ISG15-adenylate product of the adenylation reaction after the release of the PPi leaving group and Mg²⁺. With respect to the glycyl-phosphate linkage between ISG15(a) Gly157 and AMP in ISG15(a), the

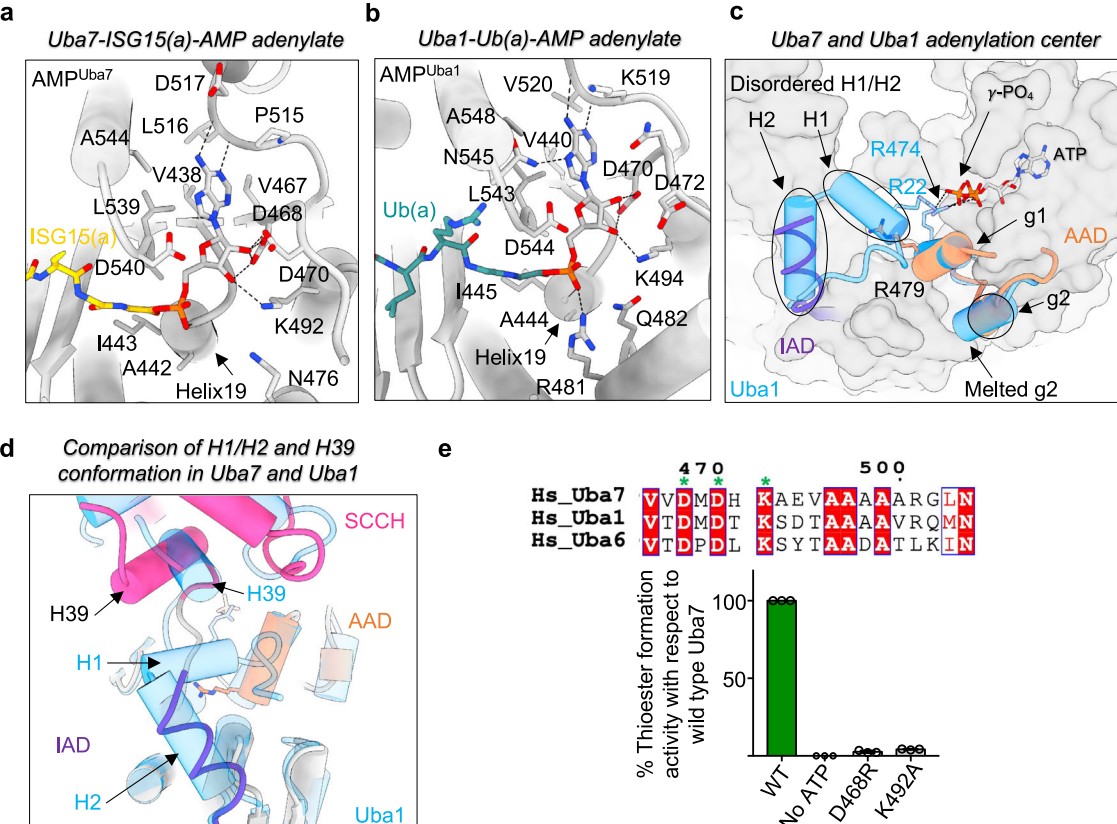

**Fig. 2 | The Uba7 active site. a** Overview of the Uba7 active site bound to ISG15-adenylate intermediate. Uba7 is shown as gray cartoon representation with residues involved in contacts to ISG15-adenylate shown as sticks. The C-terminal diglycine motif of ISG15-adenylate (yellow) is shown as sticks. **b** Overview of the Uba1 active site bound to Ub-adenylate intermediate (PDB ID: 4NNJ) presented as in panel a. **c** Superimposition of Uba1 (Cyan, PDB ID: 4II3) in the open, adenylation-competent conformation onto Uba7 from the Uba7-UBE2L6-ISG15(t)/ISG15(a) complex. The superposition highlights partial disassembly of the Uba7 active site with a disordered H1/H2 region in the IAD domain and the g1/g2 helices of the AAD extending away from the active site. Arg479 of Uba7, which is shown as sticks, projects away from the active site. **d** Similar view as c, highlighting the loss of contacts between the H1/H2 region of the IAD and the SCCH domain in the Uba7-UBE2L6-ISG15(t)/ISG15(a) complex. **e** Uba7-ISG15 thioester formation assays of the indicated mutants within the Uba7 active site. Data are represented by mean ± SEM with three independent technical replicates labeled above and individual replicates shown as black circles. The source data for **e** is provided in the source data file.

carbonyl oxygen of Gly157 and the α-phosphate of ISG15(a), which are nucleophilically attacked during adenylation and thiolation, point directly towards the *N*-terminal end of helix H19 of Uba7 AAD where they engage in hydrogen bonds with the backbone nitrogen atoms of Ala442 and Ile443, respectively (Fig. 2a). Comparison to other canonical E1 structures suggests a conserved mechanism in which the *N*-terminus of H19 represents the oxyanion hole of the active site by providing complementary positive electrostatic potential for stabilization of the transition state and tetrahedral intermediate formed during both adenylation and thioester bond formation (Fig. 2b).

Previous Ubl E1 structures have revealed that conformational coupling of the adenylation and SCCH domains plays an important role in the catalytic mechanism of E1s through its role in remodeling the active site and toggling its ability to catalyze adenylation and thioester bond formation[24,25,32,33,37]. In adenylation competent structures of Ubl E1s, the SCCH domain is perched on the H1/H2 region of the IAD with the catalytic cysteine far from the active site (Fig. 2c, d; Supplementary Fig. 4a). The H1/H2 region of the IAD and the g1/g2 region of the AAD are well-ordered and residues from these regions participate in hydrogen bonds to the β and γ-phosphates of ATP during adenylation[24,25,32,33,37] (Supplementary Fig. 4a). These contacts are thought to promote catalysis of ISG15 activation by stabilizing the pyrophosphate leaving group during adenylation, and consistent with this, mutation of H1/H2 and g1/g2 residues which contact the β and γ-phosphates of ATP results in a severe impairment of adenylation activity. Post adenylation, pyrophosphate is released and the H1/H2

and g1/g2 regions are displaced from the active site facilitating rotation of the SCCH domain which transits the catalytic cysteine into the active site for catalysis of thioester bond formation in the thiolation-competent conformation[24,25,32,33,37] (Supplementary Fig. 4b).

Interestingly, the active site of the Uba7-UBE2L6-ISG15(t)/ISG15(a) complex represents a hybrid of the adenylation-competent open and thiolation-competent closed structures (Supplementary Fig. 4c). Similar to thiolation-competent structures, the g1/g2 region of the AAD extends away from the active site and the H1/H2 of the IAD region is disordered (Fig. 2c). This displaces conserved catalytically important residues Arg17 (H1/H2) and Arg479 (g1/g2) of Uba7 from the active site (Fig. 2c). The equivalent residues of other E1s participate in contacts to the γ-phosphate of ATP during adenylation and their mutation results in a significant decrease in activity. Similar to adenylation-competent structures the SCCH domain of Uba7 adopts an open conformation despite loss of contacts to the IAD resulting from disorder of the H1/H2 region (Fig. 2d). Along these lines, we hypothesize that the comparatively poor ordering of the SCCH domain in our structures results from the loss of contacts to H1/H2 of the IAD and that the SCCH nevertheless adopts an open conformation due to interaction with UBE2L6-ISG15(t) and because ISG15(a) sterically blocks the SCCH domain from adopting a closed conformation. Partial disassembly of the Uba7 active site along with ISG15(a) promoting a conformationally plastic open conformation of the SCCH domain provides a potential molecular basis for why double loaded E1s are most efficient at catalyzing E1-E2-Ubl thioester transfer[35]. Conserved residues Asp468 and Lys492 engage in

contacts to the OH-group of the ribose sugar present in AMP of ISG15(a). Substitution of K492A and D468R results in a severe loss in ISG15 activation by Uba7, highlighting the importance of these residues for Uba7 activity (Fig. 2e; Supplementary Fig. 5a).

**Molecular basis of Uba7 recognition and specificity for ISG15(a)**
Fidelity in the parallel E1/E2/E3 cascades for Ub and Ubls represents a crucial element in biological pathways. Of particularly importance is the role of E1 enzymes, which serve as gatekeepers of the various Ub/Ubl cascades by specifically engaging and activating their cognate E2 partners. We next sought to determine the structural basis for specific recognition and activation of ISG15 by Uba7. The CTD of ISG15 exhibits between 20–37% sequence identity with Ub/Ubls of canonical E1s, with SUMO1 and Ub harboring the lowest and highest identity to ISG15 CTD, respectively. As previously mentioned, the binding surface for the conserved ISG15(a) C-terminal domain is formed by residues found in the AAD, IAD, FCCH domain, and crossover loop of Uba7. Overall, the Uba7/ISG15(a) interaction is similar to that of Uba1/Ub with a few important differences including a - 17° rotation of ISG15(a) and a complementary 25° rotation of the Uba7 FCCH domain, relative to the human Uba1/Ub structure (PDB: 6DC6)[23] (Fig. 3a). Distinct interaction networks that result from this rotation and sequence differences play a role in determining specificity as will be described in more detail below. Lastly, the NTD of ISG15 does not engage in contacts to Uba7 adenylation domain in our structure, instead it projects upward towards the UFD. This role for the NTD of ISG15 is different from that of the NTD of FAT10 which projects towards the bottom of the E1 and participates in numerous contacts with the adenylation domain that are important for FAT10 recognition[31].

The Uba7/ISG15(a) interface is continuous and composed of three distinct networks of intermolecular interactions, defined as Interface 1–3[22,23]. Interface 1 involves residues from the lower portion of the globular β-grasp the CTD of ISG15(a), which predominantly contacts the AAD of Uba7 (Fig. 3b site I). In Uba1/Ub this network of interactions is centered on "Ile44 hydrophobic patch" of Ub comprising Leu8, Ile44, His68, and Val70, which engage in contacts with Phe320, Phe926, Phe933, and Ser937 Uba1[22,23]. Alanine substitutions of Leu8 and Ile44 of the Ub Ile44 hydrophobic patch severely diminish the ability of Ub to be activated by Uba1[41]. Strikingly, residues of ISG15(a) CTD corresponding to the Ub Ile44 patch are significantly more polar (Asn89, Thr125, Phe149, and Asn151) and participate in a distinct network of interactions with Uba7 due to both sequence variability and the aforementioned rotation of the β-grasp domain of the ISG15 CTD.

Within the 'polar patch' of the ISG15(a) CTD Asn89 participates in van der Waals interactions with Uba7 Trp568, Asn151 participates in hydrogen bond and van der Waals contacts with Uba7 Tyr885 and Tyr892, and Thr125 is making hydrophobic interactions with the I894 of Uba7. While mutation of individual residues of the ISG15 polar patch (N89A, T125A, N151A) have only a modest effect on Uba7 catalyzed activation of ISG15, a triple mutant (N89A/T125A/N151A) exhibits a severe impairment of activity (Fig. 3c; Supplementary Fig. 5b). Furthermore, mutation of Uba7 surface across from the ISG15(a) polar patch also results in severe loss of thioester transfer activity. Glu890 and Tyr892, which are present near the Asn151 and Arg87 of ISG15 play an important role in facilitating thioester transfer (Fig. 3c; Supplementary Fig. 5c). Extending from this central network of interactions, Trp123 and Pro130 of ISG15(a) participate in van der Waals contacts with Tyr885, Ile894, Tyr896, and Phe899 of the Uba7 AAD (Fig. 3b site I). These contacts are unique to Uba7/ISG15 as the residues involved are poorly conserved in Uba1/Ub (Supplementary Fig. 5d). In particular, Trp123 and Pro130 of ISG15 correspond to Arg42 and Gln49 of Ub, and their hydrophobic interaction partners Ile894, Tyr896, and Phe899 of Uba7 are Gly935, Ser937 and Leu940 of Uba1, respectively (Fig. 3d). Together, these results suggest that divergence of ISG15 and Ub at their polar and hydrophobic patches and peripheral regions, respectively,

plays an important role in the specificity of these Ubls for their cognate E1s.

Interface 2 involves contacts between residues from one of the "sides" of the β-grasp domain of the ISG15(a) CTD and the FCCH domain of Uba7. Specifically, Arg92 of ISG15(a) engages in salt bridges with the side chain of Glu255 and backbone carbonyl of Glu175 as well as van der Waals contacts with Leu177 and Tyr202 in the Uba7 FCCH domain (Fig. 3e). The R92E mutation of ISG15(a) exhibits a severe reduction in Uba7-catalyzed activation, demonstrating the importance of this network of interactions (Fig. 3f). Glu115 of ISG15(a) also participates in a sidechain-mediated hydrogen bond of Tyr202 of the FCCH domain. Interface 2 is extended through hydrogen bonds between Lys90 and Ser93 of ISG15(a) and Ala174 and Arg204 of FCCH, respectively. Mutational analysis of residues from the FCCH domain which are in close proximity to Arg92 of ISG15 also shows significant role of Glu255 and Asp207 in thioester formation (Fig. 3f; Supplementary Fig. 5b and 6a). Comparison of Uba7/ISG15(a) Interface 2 to that of Uba1/Ub(a) reveals less extensive interactions due to slightly different positioning of Ubl(a) and FCCH domains that results from the aforementioned domain rotations as well as differences in sequence (Supplementary Fig. 6b).

Interface 3 involves residues from the AAD and crossover loop of Uba7, which guides the flexible ISG15 C-terminus toward the Uba7 active site. Leu154 of ISG15(a) engages in van der Waals contacts with Trp568, Gly569, and Tyr885 while Arg155 of ISG15(a) engages van der Waals and hydrogen bond interactions with Phe542 and Glu592, respectively (Fig. 3g). The diglycine motif of ISG15(a) is pinned in place through contacts between backbone carbonyls of Gly156 and Gly157 with backbone nitrogen atoms of Thr564 and Ile443, respectively (Fig. 3g). The sequence of the flexible C-termini of ISG15 and Ub are identical ('LRLRGG') and engage in a similar network of interactions in both complexes. Thus, while Interface 3 is crucial to the conserved catalytic mechanism by which Uba7 and Uba1 activate their cognate Ubls, it is unlikely to contribute to specificity in these interactions. The interaction interface between residues Gln118, Asp120, Arg155, and Arg153 of ISG15 makes contacts with crossover loop of Uba7. The triple mutant (D120K + Q118A + R153) of ISG15 shows severe loss of thioester transfer (Fig. 3h). The mutation of Glu592 shows significant reduction in the thioester transfer activity, signifying the conservation of an acidic residue near the Arg155 (Fig. 3h; Supplementary Fig. 6c).

**Combinatorial recognition of UBE2L6 by Uba7**
In the Uba7-UBE2L6-ISG15(t)/ISG15(a) complex, UBE2L6 engages in distinct sets of interactions with the UFD, the SCCH domain, and also participates in a tripartite interaction with the crossover loop of Uba7 and ISG15(a) (Fig. 4a). At the UBE2L6/UFD interface, there is a central cluster of hydrophobic interactions being decorated with a network of hydrogen bond and salt bridge-mediated interactions at the periphery (Fig. 4b Site I). Met5 and Val8 from α-helix A (hA) of UBE2L6 participates in an intricate network of van der Waals contacts with Ile945, Leu947, Leu952, Glu993, Leu994, and Ser995 of Uba7 (Fig. 4b Site I). Lys9 of UBE2L6 participates in salt bridges with Ser995 and Asp999 of Uba7 (Fig. 4b Site I). Asp999 resides within an acidic loop of the Uba7 UFD that adopts a significantly different conformation compared to Uba1 or Uba6. Glu12 of UBE2L6 hA also participates in a salt bridge with Arg944 of the Uba7 UFD. Asp29, Asn31 and Val32 engage in backbone and side chain-mediated hydrogen bonds to Ser950, Lys962 and Leu952 of the UFD, respectively (Fig. 4b Site I). Analysis of the K9E mutation in UBE2L6 shows severe impairment in thioester transfer activity (Fig. 4c). Furthermore, and consistent with our structural observations, the triple mutant (K9E + E12K + R6D) shows almost total loss of thioester transfer activity of UBE2L6 and the N31A substitution also leads to a modest reduction in activity (Fig. 4c; Supplementary Fig. 7a).

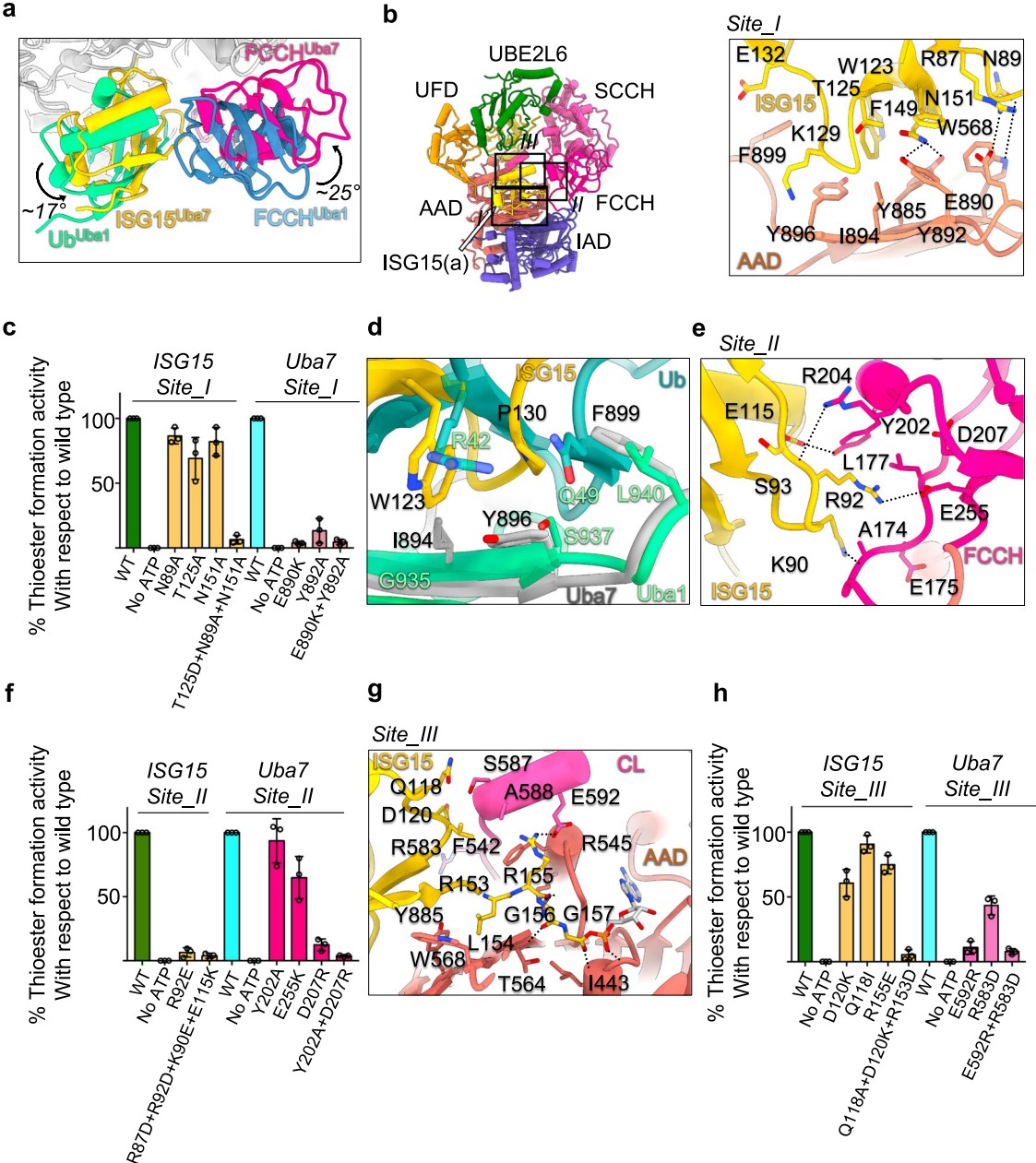

**Fig. 3 | Structural basis for Uba7/ISG15 molecular recognition. a** Relative rotations of ISG15(a) and the Uba7 FCCH domain compared to their counterparts in the Uba1/Ub(a) structure (PDB ID: 4II2). Uba7 and Uba1 were superimposed and the relevant domains/proteins are shown as cartoons and colored as labeled. **b** The Uba7-UBE2L6-ISG15(t)/ISG15(a) complex is shown as a cartoon representation (*left*) with a magnified view of Uba7/ISG15(a) interface I (right). Hydrogen bonds are indicated by dashed lines. **c** Uba7-ISG15 thioester formation assays of the indicated mutants at the Uba7/ISG15(a) site I interface. **d** Unique residues proximal to the polar patch of ISG15 (gold) participate in a distinct network of contacts to unique residues in Uba7 (gray) compared to the Uba1/Ub (spring green/cadet blue) (PDB

ID: 6DC6) interface. **e** The Uba7-UBE2L6-ISG15(t)/ISG15(a) complex structure is shown as a ribbon representation as in *panel b*, with a magnified view of Uba7/ISG15(a) interface II. **f** Uba7-ISG15 thioester formation assays of the indicated mutants at the Uba7/ISG15(a) site II interface. **g** The Uba7-UBE2L6-ISG15(t)/ISG15(a) complex structure is shown as a ribbon representation as in *panel b*, with a magnified view of Uba7/ISG15(a) interface III (CL: cross over loop). **h** Uba7-ISG15 thioester formation assays of the indicated mutants at the Uba7/ISG15(a) site II interface. Data presented in **c**, **f**, and **h** are represented by mean ± SEM with three independent technical replicates labeled above and individual replicates shown as black circles. The source data are provided in the source data file.

We also probed the role of UFD domain residues which are in contact with UBE2L6. Our functional data show the active involvement of Glu1001 and Asp999 in positioning hA of UBE2L6 onto the UFD domain to facilitate thioester transfer (Fig. 4c; Supplementary Fig. 7b). The structure of Uba7-UBE2L6-ISG15(t)/ISG15(a) complex reveals a tripartite interaction network between the UBE2L6, the crossover loop, and ISG15(a) (Fig. 4d Site II) as has been observed in other E1-E2-Ubl interactions (Fig. 4d Site II). Helix C of UBE2L6 packs against the SCCH domain of Uba7 and Glu119

and Asp127 of UBE2L6 are present within the hydrogen bonding range with Arg602 and His691 of SCCH domain (Fig. 4e Site III). E119K and D127R mutations of UBE2L6 and corresponding R602D and H691D mutations of the Uba7 SCCH domain leads to reduced thioester transfer activity (Fig. 4f; Supplementary Fig. 7a and c). This network of interactions is conserved in other structures of Uba1 in complex with Ub E2s[27–30], and is consistent with a conserved catalytic mechanism for thioester transfer among canonical Ubl E1s.

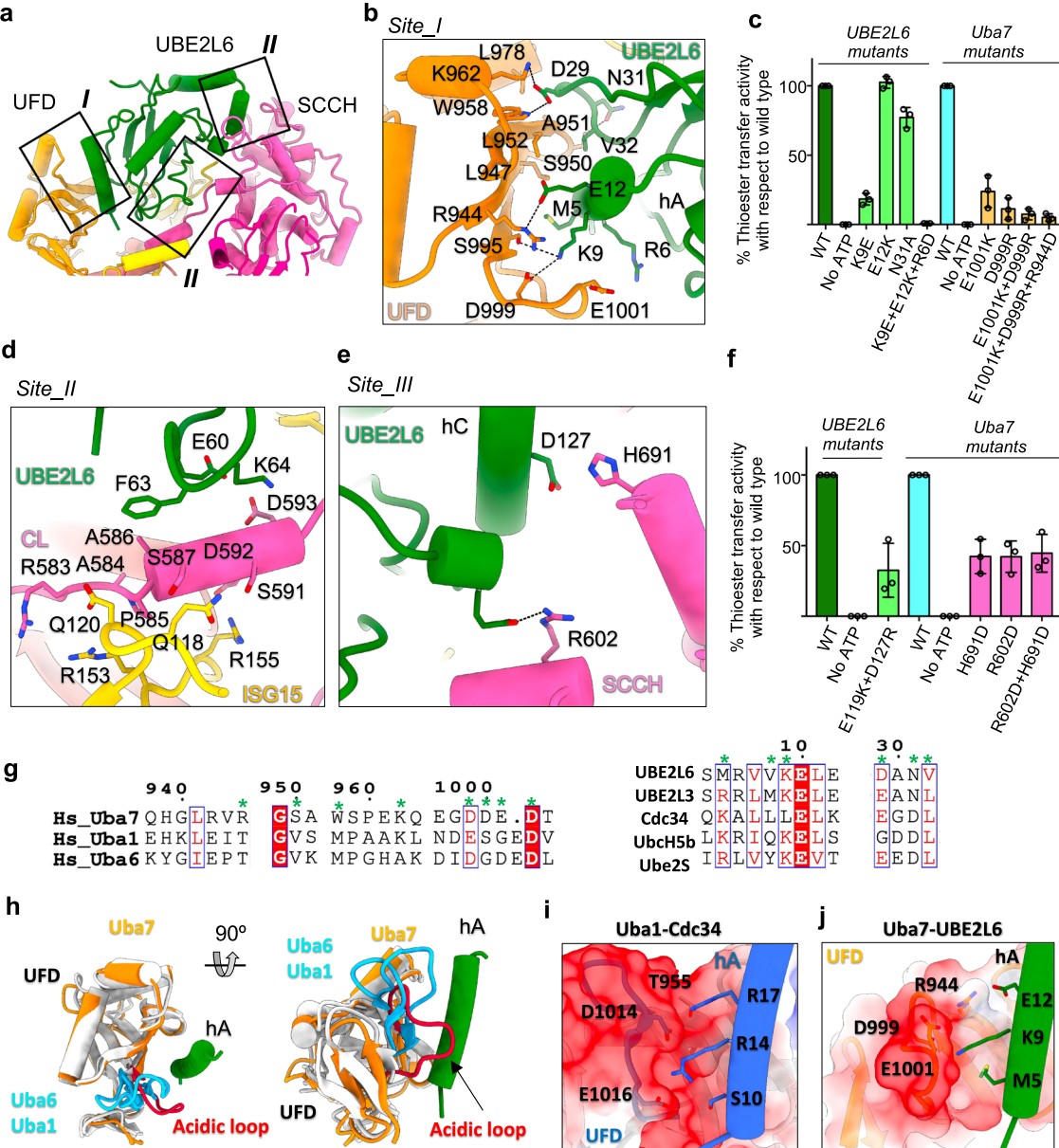

**Fig. 4 | Molecular recognition and specificity determinants of the Uba7/UBE2L6 interaction. a** The Uba7-UBE2L6-ISG15(t)/ISG15(a) complex is shown as a cartoon representation with the three distinct recognition sites for UBE2L6 boxed and labeled. **b** Magnified view of the Uba7/UBE2L6 site I interface. **c** Uba7-UBE2L6-ISG15 thioester transfer assays of the indicated mutants at the Uba7/UBE2L6 site I interface. **d**, **e** Magnified views of the Uba7/UBE2L6 site II (**d**) and site III (**e**) interfaces. **f** Uba7-UBE2L6-ISG15 thioester transfer assays of the indicated mutants at the Uba7/ UBE2L6 site III interface. **g** Sequence alignment of the indicated E1 and E2 proteins with residues involved in intermolecular interactions in the Uba7-UBE2L6-ISG15(t)/ ISG15(a) complex indicated by a green asterisk above the sequence.

**h** Superposition of the UFD of Uba7, Uba1, and Uba6 with the hA of UBE2L6 shown for reference. The acidic loop of Uba7 (red) which adopts a considerably different conformation from the corresponding loops of Uba1 and Uba6 (blue) are highlighted. **i**, **j** UFD/E2 interfaces in the Uba1-CDC34 (**i**; PDB ID: 6NYA) and Uba7-UBE2L6 (*panel j*) structures. The UFDs are shown as semitransparent surface electrostatic representations with residues contacting hA of the E2s shown as sticks. hA of the E2s are shown as cartoons, with key residues contacting the UFD shown as sticks. Data presented in **c** and **f** are represented by mean ± SEM with three independent technical replicates labeled above and individual replicates shown as black circles. The source data are provided in the source data file.

Compared to Uba1-E2 interactions in the Ub pathway, the interaction between UBE2L6 Lys9 and Ser995/Asp999 of Uba7 and between Met5 of UBE2L6 and Ile945, Leu947, Glu993, and Leu994 of the UFD are unique to the Uba7/UBE2L6 pair (Fig. 4g). A previous study revealed that a chimeric Uba7 harboring the UFD of Uba1 is unable to mediate binding of and ISG15 transfer to UBE2L6 and instead facilitates ISG15 transfer to UBE2L3, which is normally specific for Uba1 and Ub transfer[42]. The same study revealed that the Uba7-UBE2L6-ISG15 thioester transfer activity of a UBE2L6 chimera harboring the first 21 amino acids of UBE2L3 (e.g. helix A) was significantly diminished[42]. Altogether, these results suggest that the unique contacts observed at

the UFD/UBE2L6 interface in our Uba7-UBE2L6-ISG15(t)/ISG15(a) structure (Fig. 4b, c) serve as a major specificity determinant of the Uba7-UBE2L6 interaction. Moreover, the acidic loop of Uba7 shifts closer to the hA of UBE2L6 as compared to Uba1 and Uba6 (PDB code:4II2 and 7PVN) (Fig. 4h). On closer inspection we find that CDC34 has two positively charged residues (Arg14 and Arg17) in hA which engage in electrostatic interactions with the acidic loop of the Uba1 UFD (Asp1014 and Glu1016) (Fig. 4i) and UBE2L6 has only one basic residue hA (Lys9) which interacts with the acidic patch of Uba7 (Asp999 and Glu1001) (Fig. 4j). These observations further shed light on the evolution of residues at the interface between the UFD of UBA7

and hA of UBE2L6 that combinatorially ensure the fidelity of the ISG15 pathway.

## A unique ISG15(t) conformation during thioester transfer

Although modestly ordered, the ISG15(t) projects away from the surface of UBE2L6 towards the 'back' of the complex where it sits atop the AAD, sandwiched between the UFD and SCCH domain (Fig. 5a). This result was somewhat unexpected based on a previous structure of Uba1/CDC34-Ub(t) in which two distinct 'open' and 'closed' conformations of Ub(t) were observed that both reside on the front side of the complex (Fig. 5b)[27]. To validate that the density on the backside of the Uba7-UBE2L6-ISG15(t)/ISG15(a) complex indeed belongs to ISG15(t), we solved the cryo-EM structure single-loaded Uba7-UBE2L6/ISG15(a) complex lacking ISG15(t), with a nominal resolution of 3.24 Å

(masked at 0.143 FSC) (Fig. 5c; Supplementary Fig. 8; Supplementary Table S1). Analysis of the cryo-EM density reveals no density in the back of the complex that corresponds to the density present in the double-loaded Uba7-UBE2L6-ISG15(t)/ISG15(a) complex structure we ascribe to the ISG15(t) CTD (Fig. 5c).

We next sought to understand the basis for the different location of ISG15(t) and potential functional implications. As noted above, a previous study of Uba1/Cdc34-Ub(t) detected two distinct architectures of the complex: one in which the Ub thioester (Ub(t)) contacts FCCH of E1 in an open conformation and another in which Ub(t) contacts E2 in a closed conformation[27]. These two conformational states represent snapshots of the in pre- and post-thioester transfer states, and disruption of the closed conformation significantly diminished thioester transfer activity which lead to a model in which catalysis is

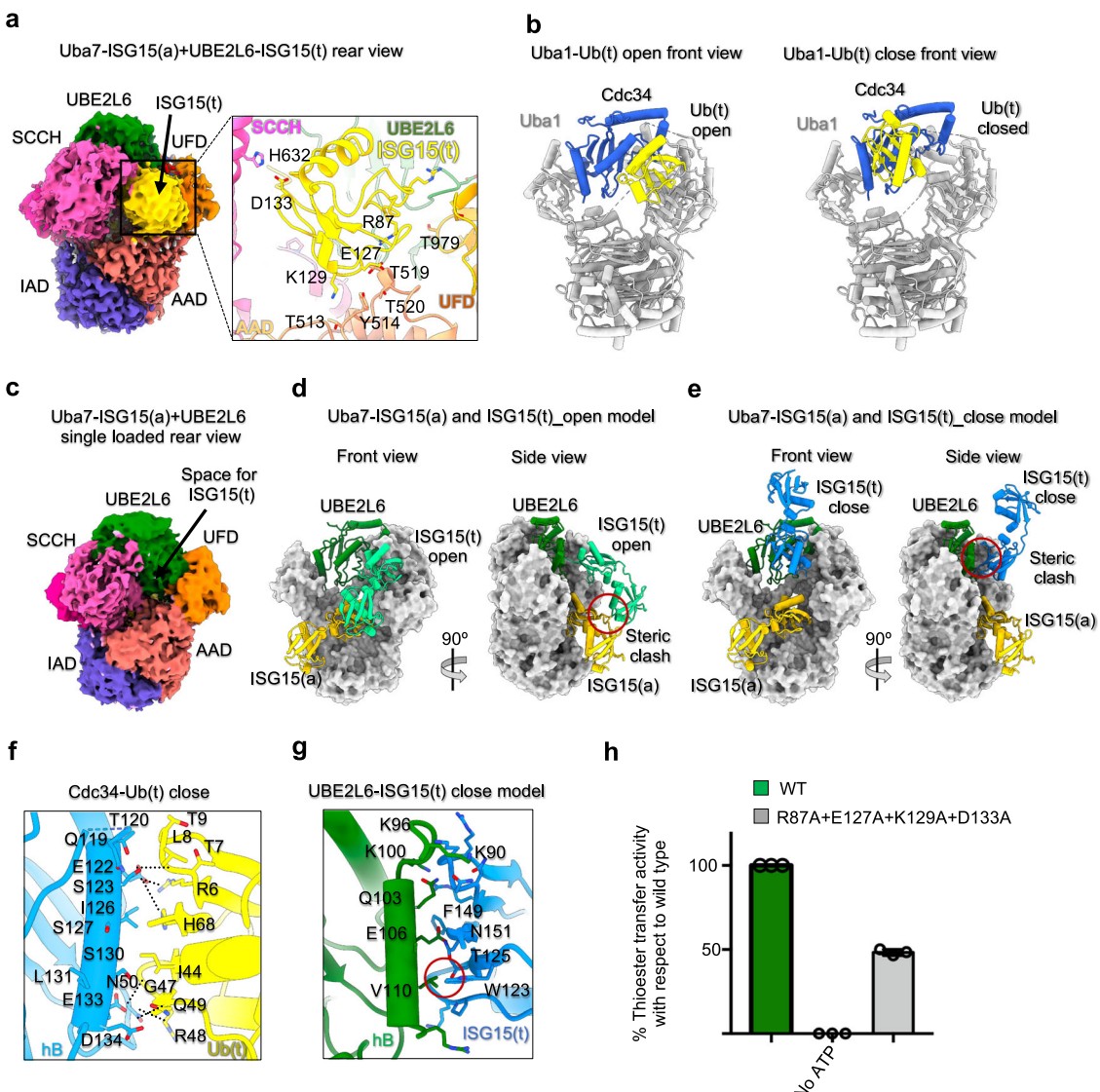

**Fig. 5 | Distinct architecture of double-loaded Uba7/UBE2L6/ISG15 compared to the Ub pathway. a** Cryo-EM map of the Uba7-UBE2L6-ISG15(t)/ISG15(a) complex (presented in a rear view) shows the presence of ISG15(t) sandwiched between the AAD, SCCH, and UFD of Uba7 (left). Docked model of ISG15(t) into the cryo-EM density with residues predicted to be in proximity of each other for possible intermolecular contacts shown as sticks (right). **b** Crystal structures of Uba1-CDC34-Ub(t) with Ub(t) in open and closed conformations shown as cartoon representations (PDB ID: 7K5J). Note that the structures in **b** are presented in a front view whereas *panel a* is presented in a rear view. **c** Cryo-EM density of the single-loaded Uba7-UBE2L6/ISG15(a) complex presented as in **a**. Note the lack of density

corresponding to ISG15(t) observed in the double-loaded complex. **d**, **e** Models of the ISG15(t) in open (**d**) and closed (**e**) conformations based on Uba1-Cdc34-Ub(t) structures presented in **b**. Steric clashes are highlighted with red circles. **f**, **g** Comparison of contacts at the closed interfaces of the CDC34-Ub(t) structure (**f**) and in the UBE2L6-ISG15(t) model (**g**). **h** Uba7-UBE2L6-ISG15 thioester transfer assays of the indicated mutants of ISG15 residues in proximity to Uba7 in the Uba7-UBE2L6-ISG15(t)/ISG15(a) structure. Data presented in **h** is represented by mean ± SEM with three independent technical replicates labeled above and individual replicates shown as black circles. The source data are provided in the source data file.

enhanced by a Ub(t)-mediated affinity switch that drives the reaction forward by promoting productive complex formation or product release depending on the conformational state. Based on our FL ISG15(a) model which is similar to previous structures of FL ISG15 alone (PDB: 1Z2M)[43] and in complex with proteases (e.g. PDB 3PSE)[44], we next docked UBE2L6 and ISG15(t) onto CDC34 and Ub(t) from the open and closed Uba1-CDC34-Ub(t) structures (Fig. 5d and e). The results of this modeling show that while the ISG15(t) CTD is compatible with the open conformation, that ISG15(t) NTD clashes with ISG15(a). Moreover, there are incompatibilities with UBE2L6-ISG15(t) adopting a closed conformation. Although there are slight differences in conformation among the many E2-Ub(t) complexes solved to date, residues from the Ile44 patch of Ub participate in a network of contacts to residues from the crossover helix on the front of E2 (e.g. hB) in all of them[28,34,45-49]. Mutation of these contacts leads to significant decrease in RING-catalyzed Ub discharge from E2 as well as a decrease in E1-E2 thioester transfer[27,34,40] (Fig. 5f).

Our UBE2L6/ISG15(t) model based on the closed Uba1-CDC34-Ub(t) structure, reveals that the divergence of hydrophobic residues from the region of ISG15 corresponding to the Ile44 patch of Ub leads to a loss of contacts with the crossover helix of UBE2L6 and also clashes between Trp123 of ISG15 and Val110 of UBE2L6. The loss of these contacts and clashes with Trp123 are also observed when UBE2L6/ISG15(t) is modeled in the closed conformation based UbcH5b-Ub(t) closed structures (Fig. 5g). Altogether, these observations provide a potential mechanism for why ISG15(t) resides on in back of the Uba7-UBE2L6-ISG15(t)/ISG15(a) structure rather than the front as in the Ub system. Given the importance of the closed E2-Ub(t) conformation for E1-E2-Ub thioester transfer[27], the failure of the UBE2L6-ISG15(t) complex to adopt a closed conformation may also contribute to previous studies observation that Uba7-UBE2L6-ISG15 thioester transfer is considerably slower than Uba1-E2-Ub thioester transfer[42,50]. With that said, the unique sandwiching of ISG15(t) in between the UFD and SCCH domain in the back of our Uba7-UBE2L6-ISG15(t)/ISG15(a) structure

suggests that these contacts may promote the open conformation of the SCCH domain and the proximal conformation of the UFD which are required for productive E1-E2 thioester transfer. Analysis of the structure reveals potential interaction surfaces of ISG15(t) with Uba7 involving ISG15 residues Arg87, Glu127, Lys129 and Asp133. Consistent with this, mutation of these ISG15 residues leads to a moderate decrease in thioester transfer activity (Fig. 5h; Supplementary Fig. 9a).

## Key determinants of ISGylation in human cells

We next examined the functional importance of key interactions observed in our Uba7-UBE2L6-ISG15(t)/ISG15(a) structure for ISGylation in human cells. Using the results of our biochemical analysis as a guide, we selected Uba7 and ISG15 mutants at the Uba7/ISG15 interface and Uba7 and UBE2L6 residues at the Uba7/UBE2L6 interface and tested the importance of these residues for both global ISGylation as well as for specific ISGylation of the viral RNA sensor MDA5 in HEK293T cells (Fig. 6a, b). The results of these assays reveal the crucial role of ISG15/Uba7 AAD and Uba7/Uba7 FCCH domain interactions for global cellular ISGylation and MDA5 ISGylation, highlighting the particular importance of Arg92 and the polar patch of ISG15 (Asn89/Thr125/Asn151) and their interacting residues on Uba7 including Glu890 and Tyr892 (Fig. 6a, b). The results also reveal the importance of interactions at the UBE2L6/UFD interface as crucial for ISGylation in cells, particularly UBE2L6 residues Lys9 and Glu12 on helix A and their interacting residues Arg944, Asp999, andGlu1001 on the Uba7 UFD (Fig. 6a, b). Notably, several Uba7, ISG15, and UBE2L6 residues we demonstrate are crucial for ISGylation activity in vitro and in cells are not conserved in their E1, E2, and Ubl counterparts in other Ubl systems. This suggests a co-evolution of Uba7, ISG15, and UBE2L6 to acquire changes that both prevent activation and thioester transfer to noncognate E1s and E2s as well as play a crucial role in Uba7-UBE2L6-ISG15 complex formation and function biochemically and in cells.

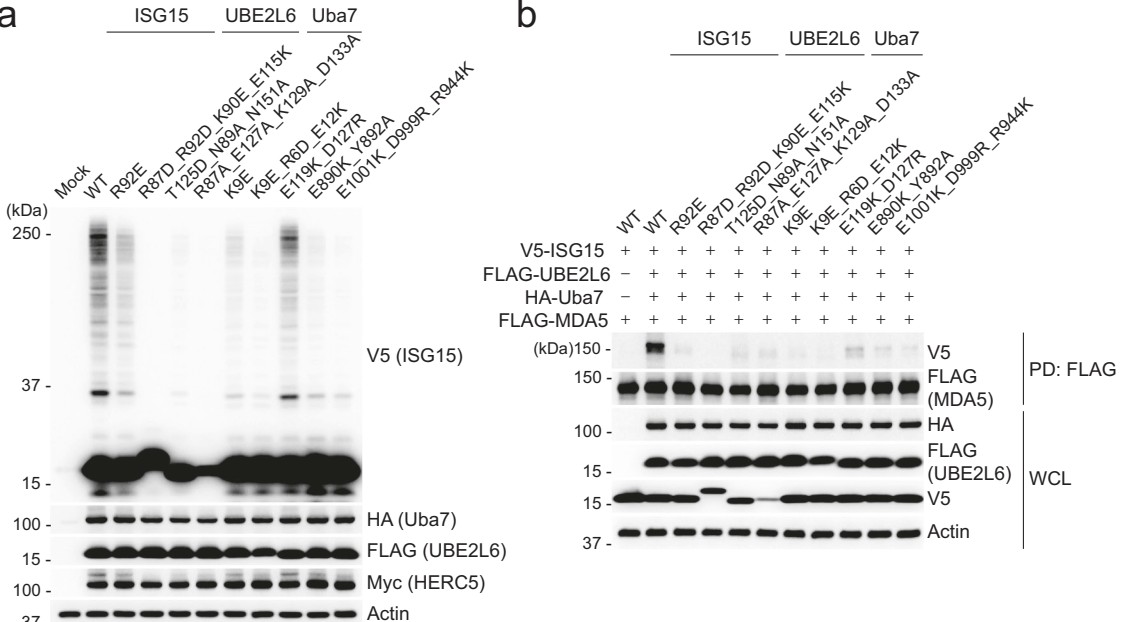

**Fig. 6 | Key determinants of the Uba7-UBE2L6-ISG15 interaction are important for ISG15 function in cells. a** Global protein ISGylation in HEK293T cells that were either transfected with empty vectors (mock) or transfected for 40 h with the set of WT proteins of the ISGylation machinery (i.e., WT HA–Uba7, FLAG–UBE2L6, Myc–HERC5 and V5–ISG15) or the indicated mutant versions of Uba7/UBE2L6/ISG15, determined in the whole-cell lysates (WCLs) by immunoblotting (IB) with

anti-V5. **b** ISGylation of FLAG–MDA5 (G74A/W75A) in HEK293T cells that were co-transfected as indicated with WT or mutant HA–Uba7, FLAG–UBE2L6, and V5–ISG15, determined by FLAG pulldown and IB with anti-V5. Data in *panels a and b* are representative of at least two independent experiments. The source data are provided as source data file.

## Discussion

In this manuscript, we have presented cryo-EM structures of human Uba7 in complex with UBE2L6, ISG15 adenylate, and ISG15 thioester intermediate in a conformational state that is poised for catalysis of Uba7-UBE2L6-ISG15 thioester transfer. Our structures reveal the basis for Uba7 recognition of ISG15 and provide insights into the molecular rules governing specificity for Uba7 over other canonical E1s, including Uba1 and Ub which have the highest sequence identity to Uba7 and ISG15, respectively. We also find divergence of the Ile44 hydrophobic patch of Ub to a polar patch in ISG15 and acquisition of unique hydrophobic residues that extend the periphery of this patch in the latter. These features of ISG15 help determine specificity for Uba7 and may also enable UBE2L6-ISG15(t) to achieve the closed conformation that is important for both E3-mediated Ub discharge and E1-E2 thioester transfer in the Ub system. Although weakly ordered in our structure, we can clearly see that the NTD of ISG15(a) does not engage in contacts to Uba7 AAD in our structure. Instead, it projects upward towards the UFD where it may promote the proximal conformation of the UFD that is required for E1-E2-ISG15 thioester transfer. The poor resolution of the ISG15(a) NTD precludes our ability to discuss potential contacts with the UFD, however, 3D variability analysis demonstrates some degree of crosstalk between these domains (Supplementary Movie 1). A potential role for the ISG15 NTD/UFD crosstalk in Uba7-UBE2L6-ISG15 thioester transfer will be the focus of future investigation. Lastly, we note that the role of the ISG15 NTD is very different from that of the NTD of FAT10 which projects towards the bottom of E1 and participates in numerous contacts with the adenylation domain that are important for FAT10 recognition[31].

Our study also provides insights into the molecular basis for UBE2L6 recognition by Uba7. Similar to E1-E2 complexes in the Ub system, UBE2L6 recognition is combinatorial and involves UFD/E2, SCCH domain/E2, and tripartite E1/E2/ISG15(a) interactions. Together with previous studies, we narrow down the specificity determinants of the Uba7/UBE2L6 pair to interactions between hA of UBE2L6 and the UFD of Uba7, which involve unique sets of residues compared to Uba1 UFD and E2s in the Ub pathway. To our surprise, ISG15(t) is located at the back of the Uba7-UBE2L6-ISG15(t)/ISG15(a) complex where it sits atop the AAD and packs against the UFD and SCCH domains. We suggest that contacts between ISG15(t) CTD and the SCCH and UFD as well as between the NTD of ISG15(a) and the UFD promote the active conformations of the SCCH and UFD to facilitate thioester transfer. We note that our biochemical data are consistent with this premise. With respect to the ISG15(a) NTD/UFD interaction, these contacts may serve as a sensor for the presence of ISG15(a) in double-loaded Uba7/UBE2L6/ISG15 complex and provide an additional mechanism as to why double-loaded complex is more efficient at thioester transfer compared to the single-loaded counterpart.

Lastly, our results show that while the SCCH domain of Uba7 is highly dynamic, it clearly interacts with UBE2L6 in a manner akin to other Uba1-E2 interactions, thus revealing a conserved catalytic mechanism among canonical E1s. The basis for this dynamism appears to be due to partial disassembly of the adenylation active site in the double-loaded structure. This partial disassembly of the Uba7 adenylation active site includes disorder of the H1/H2 region which the SCCH domain typically perches upon in other canonical E1 structures to adopt the open conformation. Loss of these contacts is a salient mechanistic feature of thioester bond formation in part by promoting conformational flexibility of the SCCH domain required to achieve the thiolation-competent closed conformation. Why such conformational dynamism of the SCCH domain was not evident in Uba1-E2 structures in the Ub pathway is unclear, and future studies will clarify whether these observations reflect real mechanistic differences or stem from X-ray crystallography being used in the determination of other E1-E2 structures.

## Methods

### Gene cloning

For protein expression in *E. coli* BL21 (codon plus), the gene fragment encoding the human Uba7 was synthesized from Gene universal into pSMT3.4 expression vector under the NdeI/EcoRI restriction sites with SMT3 tag, which can be cleaved in the presence of Ulp1 protease. The gene fragment for human UBE2L6 and ISG15 was also synthesized from Gene universal in pET29NTEV vector under the BamHI/NotI and NdeI/NotI restriction sites, respectively. The 6XHis-tag was cleaved in the presence of TEV-protease when required. For Insect cell expression, the protein encoding the hUba7 was synthesized from Gene universal in pFAST-bac HTB vector with TEV protease cleavable N-terminal 6X-His tag under the BamHI/EcoRI restriction sites. The point mutations were generated by using the PCR based site directed mutagenesis by using the primers as listed in Supplementary Table 2. The ISG15 constructs encoding the R87D + R92D + K90E + E115K, T125D + N89A + N151A, and Q118A + D120K + R153D were synthesized from Gene universal in pET29NTEV vector under the BamHI/NotI restriction sites. The K100E, K9E + E12K + R6D and E119K + D127R of UBE2L6 were also synthesized from the Gene Universal in pET29NTEV vector under the BamHI/NotI restriction sites. The E890K + Y892A, E1001K + D999R and E1001K + D999R + R944D mutant versions of Uba7 were synthesized as gBlocks from Integrated DNA technologies. For cell-based assays, pCAGGS–HA–Uba7, pFLAG–CMV2–UBE2L6, pCAGGS–V5–ISG15, and pcDNA–3 × FLAG–MDA5 (G74A/W75A) have been described previously (PMID: 33727702). pcDNA–3 × Myc–HERC5 was cloned by ligating a synthetic gBlocks gene fragment (IDT) containing the HERC5 ORF into pcDNA3.1/3 × Myc between *Nhe*I and *Not*I. The mutant versions of Uba7, UBE2L6, and ISG15 were all synthesized as gBlocks and ligated into the respective vectors used for the WT constructs.

### Protein expression and purification

All the proteins were expressed in *E. coli* BL21 codon plus (Agilent; Cat. No. 230280) as previously described[27,51]. The large scale cultures were grown until the absorbance at 600 nm was 1.5 O.D, and then cells were allowed to cool down to 18 °C. After the temperature was reached, the protein expression was induced by adding 0.3 mM isopropyl–D-1-thioglactioside (IPTG) to the final volume concentration of the culture. By utilizing the Bac-to-Bac Baculovirus Expression System, human Uba7 was expressed in insect cells. Recombinant baculoviruses with a high titer were used to infect BTI-Tn-5B1–4 (Hi5) cells (ThermoFisher Scientific; Cat. No. B85502) in Sf-900 II SFM medium (ThermoFisher Scientific) at cell density of $2 \times 10^6$ cells/ml culture. The cells were allowed to grow at 27 °C/85 rpm for 1 day and then transferred at 20 °C/85 rpm for 2 days. The cells were then harvested, and flash frozen in to liquid nitrogen and stored at −80 °C.

The bacterial or insect cells expressing the Uba7 were lysed by Continuous Flow (CF1 & CF2) Cell Disruptor at 30 psi pressure in buffer containing the 50 mM Tris-Cl pH 8.0, 300 mM NaCl, 5% Glycerol, 5 mM β-ME, 1 mM PMSF in the presence of DNaseI and lysozyme. The lysed cell supernatant was then centrifuged 39,191 × g for 45 min and then supernatant was applied to pre-equilibrated Ni-NTA resin (QIAGEN). Finally, the protein was eluted in buffer containing 20 mM Tris-Cl pH 8.0, 200 mM NaCl, 300 mM Imidazole, 5 mM β-ME. The 6X His-tag on Uba7 from insect cell expression system was cleaved in the presence of TEV protease (1/100) while the pSMT3 tag on the Uba7 protein expressed in bacteria was cleaved in the presence of Ulp1 protease (1/2000) at 4 °C overnight. The overnight cleaved protein samples were injected on the Superdex 200 26/600 PG columns (GE Healthcare) in buffer containing 20 mM Tris-Cl pH 8.0, 200 mM NaCl, 2 mM β-ME. The fractions corresponding to the protein were pooled and subjected to Enrich Q (Biorad) columns with buffer A (20 mM Tris-Cl pH 8.0, 50 mM NaCl, 2 mM β-ME) and buffer B (20 mM Tris-Cl pH 8.0, 1 M NaCl, 2 mM β-ME). The final purified fractions were concentrated up to 4 mg/ml and flash frozen into liquid nitrogen and subsequently stored at −80 °C. All mutants of Uba7

were purified using the same protocol. For ISG15, UBE2L6 and their mutants were purified by the same protocol using the same buffer compositions. We used Superdex 75 26/600 PG column to purify ISG15 and UBE2L6. The purified fractions were then concentrated up to 4–5 mg/ml and flash frozen into liquid nitrogen and stored in −80 °C.

### Uba7/ISG15 activation assay

Gel-based Uba7 thioester formation experiments were done using 100 nM Uba7, 2.5 µM ISG15, 5 mM $MgCl_2$, 2 mM ATP, 20 mM HEPES-NaOH pH 7.5, 100 mM NaCl at room temperature (RT). Reactions were started with the addition of ATP and ended by adding non-reducing urea SDS-PAGE buffer before being run on a 4–12% Mini-PROTEAN TGX Tris-glycine gel (Life Technologies) at constant 180 V for 45 min. We used the same concentrations for all the mutants of Uba7 and ISG15 under same experimental conditions. The gels were stained with Coomassie brilliant blue (Sigma-Aldrich) and imaged using a ChemiDoc MP (BioRad). Data was quantified using densitometry in ImageJ 1.53 software and processed in Prism 7.0a (GraphPad). Densitometry values on the same gel were normalized as a proportion of the control WT test. Data are represented as an average of three technical replicates with ± standard deviation error bars. Unprocessed images of representative gels for all biochemical assays are provided in the Source Data file.

### E1-E2 thioester transfer assays

Experiments were done at room temperature with 100 nM Uba7, 10 µM ISG15, 5 µM UBE2L6, 5 mM MgCl2, 2 mM ATP, 20 mM HEPES-NaOH pH 7.5, and 100 mM NaCl to form UBE2L6 ~ ISG15 thioesters. Reactions were started by adding ATP and ended by adding non-reducing urea SDS-PAGE buffer. Final reactions were then run for 45 min at a constant voltage of 180 V on a 4–12% Mini-PROTEAN TGX Tris-glycine gel from Life Technologies. All Uba7 and UBE2L6 mutants were tested under the same experimental conditions with the same concentrations. Coomassie brilliant blue (Sigma-Aldrich) was used to stain the gels, and a ChemiDoc MP was used to visualize the bands (BioRad). Data was quantified using densitometry in ImageJ 1.53 software and processed in Prism 7.0a (GraphPad). Densitometry values on the same gel were normalized as a proportion of the control WT test. Data are represented as an average of three technical replicates with ± standard deviation error bars. Unprocessed images of representative gels for all biochemical assays are provided in the Source Data file.

### Capturing a UBE2L6-ISG15 thioester mimetic

For reconstitution of the doubly loaded Uba7 ~ ISG15(a)+UBE2L6 ~ ISG15(t), first we purified the UBE2L6 (C98S/C102S/L121K) by removing all cysteines except active site C86. Because the thioester bond is highly labile, we mutated L121 which is present in the loop region near the active site C86 into lysine in order to form an iso-peptide bond with increased stability which still mimics the thioester intermediate[27,52]. Then, we also mutated Cys76 in ISG15 to avoid nonspecific crosslinking. Finally, we performed the transfer of 20 µM ISG15 C76S on the 10 µM of UBE2L6 (C98S/C102S/L121K) in the presence of 0.5 µM Uba7 in buffer 50 mM Tris-Cl pH 9.3, 50 mM NaCl, 10 mM $MgCl_2$, 2 mM DTT and 2 mM ATP after overnight incubation at 37 °C. The reaction mixture was then passed over pre-equilibrated Ni-NTA beads to remove the excess UBE2L6 (C98S/C102S/L121K). Then UBE2L6 (C98S/C102S/L121K) ~ ISG15 (C76S) (hereafter UBE2L6 ~ ISG15(t)) was purified using EnrichQ (BioRad) anion exchange with buffer A (20 mM Tris-Cl pH 8.0, 50 mM NaCl) and buffer B (20 mM Tris-Cl pH 8.0, 1 M NaCl). The final purified fraction was concentrated up to 3 mg/ml and flash frozen in liquid nitrogen and stored at −80 °C.

### Trapping the Uba7 ~ ISG15(a)+UBE2L6 ~ ISG15(t) doubly-loaded complex

The active site Cys86 of purified UBE2L6 ~ ISG15(t) were activated in the presence of 5 mM Aldrithiol™-4 (ADT) (Sigma-Aldrich) and excess ADT was removed by desalting in buffer A (20 mM Tris-Cl pH 8.0, 50 mM NaCl). This activated UBE2L6 - ISG15(t)-ADT was then incubated with purified Uba7 in a 1:1 stochiometric ratio for 45 min at room temperature in buffer A (20 mM Tris-Cl pH 8.0, 50 mM NaCl). Finally, we purified the Uba7 + UBE2L6 ~ ISG15(t) and removed free Uba7 and UBE2L6 ~ ISG15(t)-ADT with EnrichQ anion exchange in buffer A (20 mM HEPES pH 7.5, 50 mM NaCl) and buffer B (20 mM HEPES pH 7.5, 1 M NaCl). The purified fractions corresponding to the Uba7 + UBE2L6 ~ ISG15(t) were then incubated with 1.2 molar excess of ISG15, ATP and $Mg^{2+}$ for 30 min at 4 °C. Finally, the sample was concentrated to 3 mg/ml for grid preparation. For Uba7 ~ ISG15(a)+UBE2L6 single loaded complex, we used the same strategy as for double loaded complex with minor change in the use to UBE2L6 instead of UBE2L6 ~ ISG15(t).

### Cell culture and plasmid transfection

HEK293T (CRL-3216) cells were purchased from American Type Culture Collection (ATCC) and were maintained in Dulbecco's modified Eagle's medium (DMEM, Gibco) supplemented with 10% (v:v) fetal bovine serum (FBS, Gibco), 2 mM Glutamine (Gibco), 1 mM sodium pyruvate (Gibco) and 100 U/ml of penicillin–streptomycin (Gibco). Transient DNA transfections were performed using *Trans*IT-LT1 Transfection Reagent (Mirus) as per the manufacturers' instructions.

### Immunoprecipitation and immunoblotting

HEK293T cells that were transfected for 40 h with FLAG-MDA5 (G74A/W75A) together with WT or mutant ISGylation machinery components were lysed in Nonidet P-40 (NP-40) buffer (50 mM Tris-HCl, pH 7.5, 200 mM NaCl, 1% (v:v) IGEPAL® CA-630, 1 mM ethylenediaminetetraacetic acid (EDTA) and 1 × protease inhibitor cocktail (MilliporeSigma)). The cell lysates were cleared by centrifugation at $20,000 \times g$ and 4 °C for 20 min and then subjected to FLAG pulldown at 4 °C for 16 h using Dynabeads Protein G (Thermo Fisher Scientific) pre-conjugated with the anti-FLAG M2 antibody (MilliporeSigma). The beads were extensively washed with NP-40 buffer and proteins were eluted by heating in 1 × Laemmli SDS sample buffer at 95 °C for 5 min. Protein samples were resolved on Bis–Tris SDS-polyacrylamide gel electrophoresis (PAGE) gels, transferred onto polyvinylidene difluoride (PVDF) membranes (Bio-Rad), and visualized using the SuperSignal West Pico PLUS chemiluminescence reagents (Thermo Fisher Scientific) on an ImageQuant LAS 4000 Chemiluminescent Image Analyzer (General Electric) as previously described[7]. The antibodies used for immunoblotting include: anti-V5 (1:1,000, SV5-Pk1; Invitrogen), anti-HA (1:1,000, C29F4; Cell Signaling Technology), anti-FLAG (1:1,000, M2; MilliporeSigma), anti-Myc (1:1,000, 9B11; Cell Signaling Technology), and anti-β-Actin (1:500, C4; Santa Cruz).

### Cryo-EM sample preparation and data collection

3ul of freshly purified respective complexes were applied on to the UltrAuFoil 1.2/1.3 300 mesh grids that had been glow discharged at 20 mA for 30 s in a Quorum EMS glow discharger. Grids were vitrified with a Vitrobot Mark IV (Thermo Fisher Scientific) maintained at 8 °C and 100% humidity. A blot force of 0 and blot time of 3 s were applied before plunge freezing into liquid ethane. The Uba7 ~ ISG15(a)+UBE2L6 ~ ISG15(t) doubly loaded complex was frozen at a concentration of 3.0 mg/ml while the Uba7 ~ ISG15(a)+UBE2L6 single loaded complex was frozen at 2.8 mg/ml. We collected data of Uba7 ~ ISG15(a)+UBE2L6 ~ ISG15(t) doubly-loaded complex using a Titan Krios G3 microscope (ThermoFisher Scientific) operating at 300 kV at UT Austin, Texas with Gatan Biocontinuum Imaging Filter and a K3 direct electron detector Gif+DED. We collected 14011 movies with 0.8332 Å pixel size at a nominal 130,000 TEM magnification. The total dose of 80 $e^-/Å^2$ with 125 frames and −800 to −2500 nm defocus range was utilized during data collection. For the Uba7 ~ ISG15(a)+UBE2L6 single loaded complex, we collected data on G3i Cryo Transmission Electron

Microscope at Empire High-end EM at LBMS, Brookhaven National Laboratory, New York. We collected 9831 movies using super resolution model with a pixel size 0.415 Å at a nominal 130,000 TEM magnification. The total dose of 53 e$^-$/Å$^2$ with 45 frames and −800 to −2500 nm defocus range was utilized during data collection.

## Data processing
The final images were processed in CryoSPARC 3.1[53]. Motion correction was performed by using the MotionCor2 and contrast transfer function was estimated using patch CTF estimation (multi) in cryoSPARC 3.1. For both datasets the particles were picked by using the blob picker from only 500 micrographs and templates were prepared. Using good class averages as templates, we began picking particles from the whole dataset. The particle picking was then inspected and extracted. Several rounds of 2D classification were performed to remove the junk particles Ab initio reconstructions were then utilized to perform heterogenous refinement. The particle stacks which were showing density for all the components were further subjected to 3D classification. Then non-uniform refinement was performed for each 3D map. Due to the weak density of the SSCH domain throughout most classes, focused 3D classification was applied to improve the resolution within the SCCH domain. This approach resulted in distinguishable density for all components of the complex in two classes and subsequent non-uniform refinements resulted in considerable improvement of SCCH domain density. Furthermore, we have also performed the 3D variability analysis to identify the distinct and continuous heterogeneity in the single particles of Uba7 - ISG15(a)+UBE2L6 - ISG15(t) complex in CryoSPARC 3.1.

## Model building and refinements
The AlphaFold models of Uba7 and UBE2L6 and ISG15 were used to initially dock the models onto the maps of Uba7 - ISG15(a)+UBE2L6 - ISG15(t) and Uba7 - ISG15(a)+UBE2L6 complex using the ChimeraX-1.3[54] and coordinates were generated for the complex. The real space refinement module in Phenix-1.20.1[55] was used to perform the refinements in the respective maps. Coordinates and maps were manually inspected in Coot 0.9.8.7[56] for rotamer outlier, clashes and Ramachandran outliers. Iterative rounds of real space refinement and model building in Coot 0.9.8.7 was conducted to generate optimized coordinates for the complex.

## Reporting summary
Further information on research design is available in the Nature Portfolio Reporting Summary linked to this article.

## Data availability
Atomic coordinates for the Uba7/UBE2L6/ISG15 complexes reported in this study are deposited in the RCSB with accession codes 8SE9 (double-loaded, Form 2), 8SEA (double-loaded, Form 1), 8SEB (single-loaded), and 8SV8 (double-loaded composite) and the corresponding cryo-EM maps have been deposited into the Electron Microscopy Data Bank with accession numbers EMD-40407, EMD-40408, EMD-40409, EMD-40782, and consensus map as EMD-40799. The structural data used from RCSB are listed below: PDB: 4NNJ, PDB:4II3, PDB:6DC6, PDB:6NYA, PDB:7K5J, PDB:4II2, PDB:6NYO, PDB:6O83. Source data are provided with this paper.

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

## Acknowledgements

The authors thank member of the Olsen and Wasmuth Laboratories for helpful discussions. Research reported in this publication was supported by NIH grants R01 GM115568 and R01 GM128731 (S.K.O.); R01 CA168635, RO1 ES007061, PO1 CA92584, R35 CA241801 (P.S.); R00 GM140264 (E.V.W.); and R37 AI087846 (M.U.G.). P.S. is the holder of the Robert A. Welch Distinguished Chair in Chemistry (AQ-0012), recipient of a CPRIT REI Award (RR180029), and receives support from and a Gray Foundation Team Science Award under the Basser BRCA Initiative. S.K.O. is the recipient of a CPRIT Rising Star Award (RR200030), and E.V.W. is the recipient of a CPRIT Recruitment of First Time Tenure Track Faculty Award (RR220068). Cryo-EM screening was conducted at the UT Health San Antonio Cryo-EM Facility on a Glacios TEM equipped with a Falcon IV camera and Selectris energy filter which were purchased with the support of UT STARs awards 402-1288 (P.S.) and 402-1317 (S.K.O.). We thank Axel Brilot and Evan Schwartz for collecting cryo-EM data sets used for final reconstructions of double-loaded complex reported in this study at the University of Texas at Austin Sauer Structural Biology Laboratory (RRID:SCR_022951). Cryo-EM data for the single-loaded complex were collected at The Laboratory for BioMolecular Structure (LBMS), which is supported by the DOE Office of Biological and Environmental Research (KP1607011). This research utilized resources of the Structural Biology Core Facilities, part of the Institutional Research Cores at the University of Texas Health Science Center at San Antonio supported by the Office of the Vice President for Research and the Mays Cancer Center Drug Discovery and Structural Biology Shared Resource (NIH P30 CA054174). The content of this study is solely the responsibility of the authors and does not necessarily represent the official views of the NIH.

## Author contributions

Molecular cloning was conducted by M.A., X.Z., and A.S. M.A., K.E.C. and A.N. conducted insect cell expression. Protein purification was conducted by M.A. and P.D.S.B, Cryo-EM grid preparation and screening were conducted by L.J.. Data processing, model building, and analysis were conducted by M.A., L.J., E.A.R., D.N., E.V.W., and S.K.O. M.A. conducted biochemical assays with assistance from D.N. G.L. and Z.S. conducted cell-based assays under supervision of M.G.U., P.S. assisted with experimental design and data interpretation. The figures and manuscript

were prepared by M.A., E.A.R., D.N., and S.K.O., with input from all authors.

## Competing interests

The authors declare no competing interests.
