## [Peer Review File · Nature Communications]

REVIEWER COMMENTS

Reviewer #1 (Remarks to the Author):

In this work, Afsar et al provides a cryo-EM structures of human Uba7 in complex with UBE2L6, ISG15 adenylate (ISG15a), and ISG15 thioester (ISG15t) intermediate that are poised for catalysis of Uba7-UBE2L6-ISG15 thioester transfer. This work provides the molecular basis of Uba7's specificity towards ISG15 and UBE2L6. They have uniquely trapped doubly-loaded Uba7-UBE2L6-ISG15(t)/ISG15(a) mimetic complex which reveal unique conformations of the ISG15-adenylate NTD and ISG15 thioester intermediate CTD that promote the active conformation of the Uba7-UBE2L6-ISG15(t)/ISG15(a) complex. Their biochemical analysis revealed the key determinant of Uba7-catalyzed ISG15 activation and thioester transfer to UBE2L6. They have also shown that these key molecular determinants are crucial for ISG15 function in cells as mutation leads to reductions of bulk and MDA5 ISGylation in human cell-based studies. Overall, this is an important structure in the ISG15 biology and there are no major concerns. but the authors should improve the presentation of their data and writing is also patchy all along. Refer to my specific comments below.

In figure 1B and C, It would be clearer to color the labels of the individual domains and proteins in the cryo-EM density with the same color in which the domain's EM density is colored.

The two forms of the complex structures shown in Figure 1 do not seem to be functionally different, in fact there are only very minor changes in one domain of UBA7, so what warrants the presentation of two models for this and what is the comparison of these two forms telling us?

In line 156, authors text starts reading "Cryo-EM density linking Gly157....". There is no figure to support the claim made in this sentence.

In line 2013, a sentence starting with "Comparison to other canonical E1 structures....." is there. It talks about Helix H19's importance and cites figure 2B. but I could not find H19 in the figure.

In line 213, authors have a sentence starting with "In adenylation competent structures of Ubl E1s.....". This is a long sentence with E1 structural jargon g1, g2 , H1, H2 .. these regions are not introduced until this point and no figure provided to understand what the authors mean.

In line 227, it states "Similar to adenylation-competent structures the SCCH domain of Uba7 adopts an open conformation despite loss of contacts to the IAD resulting from disorder of the H1/H2 region"

-to make such comparison, one needs to provide figures for adenylation-competent structure and showcase a comparison. Or if there is already a figure, please refer to it.

In line 251, it states that "Overall, the Uba7/ISG15(a) interaction is similar to that of Uba1/Ub with a few important differences including a $\sim 17^\circ$ rotation of ISG15(a) and a complementary 25° rotation of the Uba7 FCCH domain, relative to the human Uba1/Ub structure (PDB: 6DC6) 23." , There should be a structure figure to support such a statement.

In line 363, this sentence "These results, together with our structural observations that the E2/SCCH and tripartite interfaces are largely conserved between Uba7/UBE2L6 and Uba1/E2/Ub structures suggest that the unique contacts noted above at the UFD/UBE2L6 interface observed in our structure serve as a major specificity determinant of the Uba7-

UBE2L6 interaction." Has to be rewritten.

In line 440, this sentence "Notably, several of the Uba7, ISG15, and UBE2L6 residues our results show are crucial for ISGylation in our biochemical and cell-based assays are also specific to the ISG15 pathway as noted above", should be rewritten.

In the discussion line 459), authors speculate a role for the NTD of ISG15 in position of the UFD. But this seems a bit far-fetched considering there is hardly any specific interaction. If there are specific interactions, these should be described and shown.

Reviewer #2 (Remarks to the Author):

Afsar et al have tackled outstanding questions in the ISG15 field about the activity, specificity, and mechanism of action of Uba7 and Ube2L6 recognition of ISG15 by solving several multi-protein cryo-EM structures which simulate the activation and conjugation of ISG15. This structural insight allowed them to compare and contrast Uba7 activity with other E1s for ubiquitin and FAT10 and to generate mutants which test hypotheses from the new structures. The paper provides necessary and critical structural information on this process and will be impactful for a variety of groups who study ISG15, ISGylation, and E1/E2 activation/conjugation systems. The work is well done, well controlled and exciting though I have several minor suggestions for improvement.

The writing is clear but for a non-structural biology audience and it may be useful to generate videos which model how the authors imagine the proteins to move during the E1/E2 transfer and to associate these videos with the article.

Why does the mutant of ISG15 (R87D, R92D, K90E, E115K) lead to an upshifted ISG15 signal (Figure 6A)? Does it migrate more slowly based on the mutations or is there a biological reason for the upshifted band in cells? It is difficult to estimate how large the shift is.

Minor suggestions:

Line 186 "modest" should probably be "most"

Line 440 "partners" could be "residues"

REVIEWER COMMENTS

Reviewer #1 (Remarks to the Author):

In this work, Afsar et al provides a cryo-EM structures of human Uba7 in complex with UBE2L6, ISG15 adenylate (ISG15a), and ISG15 thioester (ISG15t) intermediate that are poised for catalysis of Uba7-UBE2L6-ISG15 thioester transfer. This work provides the molecular basis of Uba7's specificity towards ISG15 and UBE2L6. They have uniquely trapped doubly-loaded Uba7-UBE2L6-ISG15(t)/ISG15(a) mimetic complex which reveal unique conformations of the ISG15-adenylate NTD and ISG15 thioester intermediate CTD that promote the active conformation of the Uba7-UBE2L6-ISG15(t)/ISG15(a) complex.

Their biochemical analysis revealed the key determinant of Uba7-catalyzed ISG15 activation and thioester transfer to UBE2L6. They have also shown that these key molecular determinants are crucial for ISG15 function in cells as mutation leads to reductions of bulk and MDA5 ISGylation in human cell-based studies. Overall, this is an important structure in the ISG15 biology and there are no major concerns. but the authors should improve the presentation of their data and writing is also patchy all along. Refer to my specific comments below.

We are grateful to the reviewer for their positive comments and thoughtful analysis of our study.

In figure 1B and C, It would be clearer to color the labels of the individual domains and proteins in the cryo-EM density with the same color in which the domain's EM density is colored.

We have colored the labels according to the reviewer's suggestion.

The two forms of the complex structures shown in Figure 1 do not seem to be functionally different, in fact there are only very minor changes in one domain of UBA7, so what warrants the presentation of two models for this and what is the comparison of these two forms telling us?

We agree with the reviewer's comment that the two forms of the complex do not seem to be functionally different. As noted in the original manuscript text, there are minor differences in the quality of the cryo-EM density for individual domains of the components of the complex between Forms 1 and 2 (e.g. the SCCH domain, ISG(t), and the NTD of ISG15(a)). Accordingly, we have generated a composite map (EMDB ID 40782) which the coordinates were subsequently refined against to create a final model (PDB: 8SV8; validation report provided). Figure 1, Supplementary Figure 2, and Supplementary Table 1 have been updated accordingly.

In line 156, authors text starts reading "Cryo-EM density linking Gly157....". There is no figure to support the claim made in this sentence.

We apologize for this omission and in Supplementary Figs. 3a and b, we now present the cryo-EM density for: 1) the disulfide bond linking the active site cysteines of Uba7 (Cys599) and UBE2L6 (Cys86) (panel a) and 2) the isopeptide bond linking Lys121 of UBE2L6 and Gly157 of ISG15(t) in the UBE2L6-ISG15(t) thioester mimetic (panel b).

In line 213, a sentence starting with “Comparison to other canonical E1 structures.....” is there. It talks about Helix H19’s importance and cites figure 2B. but I could not find H19 in the figure.

We thank the reviewer for pointing this out. We have corrected a typo and have labeled H19 in Fig. 2b more prominently in the revised manuscript.

In line 213, authors have a sentence starting with “In adenylation competent structures of Ubl E1s.....”. This is a long sentence with E1 structural jargon g1, g2 , H1, H2 .. these regions are not introduced until this point and no figure provided to understand what the authors mean.

We have defined these structural features including the roles they play in catalysis and now include a figure which compares and contrasts the catalytic states in Supplementary Fig. 4. We have also updated main Fig. 2c to better illustrate the roles of these regions in catalysis.

In line 227, it states “Similar to adenylation-competent structures the SCCH domain of Uba7 adopts an open conformation despite loss of contacts to the IAD resulting from disorder of the H1/H2 region” -to make such comparison, one needs to provide figures for adenylation-competent structure and showcase a comparison. Or if there is already a figure, please refer to it.

We thank the reviewer for highlighting this omission and a figure providing such a structural comparison is presented in new Supplementary Fig. 4.

In line 251, it states that “Overall, the Uba7/ISG15(a) interaction is similar to that of Uba1/Ub with a few important differences including a ~17° rotation of ISG15(a) and a complementary 25° rotation of the Uba7 FCCH domain, relative to the human Uba1/Ub structure (PDB: 6DC6) 23.”, There should be a structure figure to support such a statement.

We have updated and relabeled Fig. 3a to better highlight the relative rotations of the Ubls and FCCH domains.

In line 363, this sentence “These results, together with our structural observations that the E2/SCCH and tripartite interfaces are largely conserved between Uba7/UBE2L6 and Uba1/E2/Ub structures suggest that the unique contacts noted above at the UFD/UBE2L6 interface observed in our structure serve as a major specificity determinant of the Uba7-UBE2L6interaction.” Has to be rewritten.

We have rewritten this sentence for improved clarity.

In line 440, this sentence “Notably, several of the Uba7, ISG15, and UBE2L6 residues our results show are crucial for ISGylation in our biochemical and cell-based assays are also specific to the ISG15 pathway as noted above”, should be rewritten.

We have rewritten this sentence for improved clarity.

In the discussion line 459), authors speculate a role for the NTD of ISG15 in position of the UFD. But this seems a bit far-fetched considering there is hardly any specific interaction. If there are specific interactions, these should be described and shown.

We appreciate the reviewer’s comment. In the revised manuscript text, we have further highlighted that potential crosstalk between the UFD and ISG15 NTD is speculative and note that this topic will be a focus of future investigation. We also performed 3D variability analysis which we have included as Supplementary Movie 1 which we hope better highlights the basis for our speculation.

Reviewer #2 (Remarks to the Author):

Afsar et al have tackled outstanding questions in the ISG15 field about the activity, specificity, and mechanism of action of Uba7 and Ube2L6 recognition of ISG15 by solving several multi-protein cryo-EM structures which simulate the activation and conjugation of ISG15. This structural insight allowed them to compare and contrast Uba7 activity with other E1s for ubiquitin and FAT10 and to generate mutants which test hypotheses from the new structures. The paper provides necessary and critical structural information on this process and will be impactful for a variety of groups who study ISG15, ISGylation, and E1/E2 activation/conjugation systems. The work is well done, well controlled and exciting though I have several minor suggestions for improvement.

The writing is clear but for a non-structural biology audience and it may be useful to generate videos which model how the authors imagine the proteins to move during the E1/E2 transfer and to associate these videos with the article.

We thank the reviewer for this suggestion and now include Supplementary Movie 2, which helps with visualization of how the composition and architecture of the Uba7 complexes change during its catalytic cycle. We note that several of these steps depicted in the movie are based on structures of other canonical E1 enzymes such as Ub E1 and SUMO E1 as the corresponding structures of Uba7 have not yet been determined (e.g. adenylation and thioester competent).

Why does the mutant of ISG15 (R87D, R92D, K90E, E115K) lead to an upshifted ISG15 signal (Figure 6A)? Does it migrate more slowly based on the mutations or is there a biological reason for the upshifted band in cells? It is difficult to estimate how large the shift is.

We believe this shift is due to charge differences in the mutant protein relative to WT which results in altered migration on SDS-PAGE. We considered that the upshift in migration could be due to posttranslational modification of the mutant protein when expressed in human cells but we also see altered migration of some of the bacterially-derived mutants on SDS-PAGE (please see Fig. 5b).

Minor suggestions:

Line 186 “modest” should probably be “most”
We have made the recommended change.

Line 440 “partners” could be “residues”
We have made the recommended change.